# Predicting Multimorbidity Using Saudi Health Indicators (Sharik) Nationwide Data: Statistical and Machine Learning Approach

**DOI:** 10.3390/healthcare11152176

**Published:** 2023-07-31

**Authors:** Faisal Mashel Albagmi, Mehwish Hussain, Khurram Kamal, Muhammad Fahad Sheikh, Heba Yaagoub AlNujaidi, Sulaiman Bah, Nora A. Althumiri, Nasser F. BinDhim

**Affiliations:** 1College of Applied Medical Sciences, Imam Abdulrahman bin Faisal University, Dammam 31441, Saudi Arabia; fmalbagmi@iau.edu.sa; 2College of Public Health, Imam Abdulrahman bin Faisal University, Dammam 31441, Saudi Arabia; hnujaidi@iau.edu.sa (H.Y.A.); sbah@iau.edu.sa (S.B.); 3Department of Engineering Sciences, National University of Sciences and Technology, Islamabad 44000, Pakistan; khurram.kamal@pnec.nust.edu.pk; 4Department of Mechanical Engineering, University of Management and Technology, Sialkot Campus, Lahore 54770, Pakistan; muhammad.fahad@skt.umt.edu.pk; 5Sharik Association for Research and Studies, Abubaker Alsedeq, Riyadh 13326, Saudi Arabia; nora@althumiri.net (N.A.A.); nd@nasserdhim.com (N.F.B.)

**Keywords:** multimorbidity, prediction, health indicators surveillance, logistic regression, backpropagation methods

## Abstract

The Saudi population is at high risk of multimorbidity. The risk of these morbidities can be reduced by identifying common modifiable behavioural risk factors. This study uses statistical and machine learning methods to predict factors for multimorbidity in the Saudi population. Data from 23,098 Saudi residents were extracted from the “Sharik” Health Indicators Surveillance System 2021. Participants were asked about their demographics and health indicators. Binary logistic models were used to determine predictors of multimorbidity. A backpropagation neural network model was further run using the predictors from the logistic regression model. Accuracy measures were checked using training, validation, and testing data. Females and smokers had the highest likelihood of experiencing multimorbidity. Age and fruit consumption also played a significant role in predicting multimorbidity. Regarding model accuracy, both logistic regression and backpropagation algorithms yielded comparable outcomes. The backpropagation method (accuracy 80.7%) was more accurate than the logistic regression model (77%). Machine learning algorithms can be used to predict multimorbidity among adults, particularly in the Middle East region. Different testing methods later validated the common predicting factors identified in this study. These factors are helpful and can be translated by policymakers to consider improvements in the public health domain.

## 1. Introduction

In general, the coexistence of multiple diseases in an individual is referred to as multimorbidity. When a primary disease, such as diabetes, is identified, the additional diseases present are known as comorbidities of the primary disease [1]. The study of comorbidity and multimorbidity has seen significant growth over the past four decades. A comprehensive analysis of global research on comorbidity and multimorbidity revealed that between 1970 and 2016, a total of 85,994 papers were published from 168 countries [2]. One contributing factor to the increasing importance of research in this area is the aging population. As people age, the likelihood of experiencing multiple diseases rises. This trend is observed worldwide, as all regions of the world are facing the challenges of an aging population. By 2050, it is projected that one in every six individuals globally will be aged 65 or older, compared to one in every eleven in 2019 [3]. A large-scale, multi-country study confirmed the high prevalence of multimorbidity among adults aged 50 and above in low-, middle-, and high-income countries [4].

Besides examining levels, trends, and determinants, research on comorbidity and multimorbidity aims to uncover patterns and explore the etiological relationships between diseases. The aforementioned multi-country study identified and distinguished three types of multimorbidity patterns in the nine countries under investigation: “cardio-respiratory” (angina, asthma, and chronic obstructive pulmonary disease), “metabolic” (diabetes, obesity, and hypertension), and “mental-articular” (arthritis and depression) [4]. A large-scale cohort study conducted in the UK found that older patients, women, and socioeconomically deprived groups had a higher number of comorbidities. Additionally, the patterns of comorbidity varied according to age and sex [5]. Expanding on the work of Valderas et al. [1] proposed four etiological models of comorbid diseases: direct causation (where one comorbid disease causes the other), associated risk factors (when the risk factors of the comorbid diseases are correlated), heterogeneity (when the risk factors for the comorbid diseases are uncorrelated, but the risk factor for one comorbid disease is also a risk factor for the other), and lastly, independence (when the presence of diagnostic features of comorbid diseases is due to another comorbid disease).

The three primary sources of data for research on comorbidity and multimorbidity are electronic health records, surveys (either of patients or the general population), and administrative records [6]. Each source has its advantages, disadvantages, and specific tools and methods. The study by Garin et al. [4] serves as an example of research based on survey data, while the study by Tran et al. [5] exemplifies research based on electronic health records. Tonelli et al. [7] have described the methods for identifying 30 comorbidities from administrative data#. Administrative data and electronic health records often involve large datasets reaching millions of records. These datasets are increasingly being analysed using big data methods and machine learning [8,9].

In machine learning, it is stated that “a computer program is said to learn from experience E with respect to some class of tasks T and performance measure P if its performance at tasks T, as measured by P, improves with experience E” [10]. Some common machine learning tasks include classification, classification with missing inputs, regression, transcription, machine translation, structured outputs, anomaly detection, imputation of missing values, denoising, and density estimation. Regarding “experience E”, numerous algorithms are broadly grouped under unsupervised and supervised learning algorithms [11].

The Saudi population is at a high risk of various morbidities. Children, adults, and older adults of any gender are susceptible to these uncontrolled morbidities [12,13,14]. In the study by Alhabib et al. [15], which involved 2047 participants, approximately 30.3% had hypertension, 25.1% had diabetes, 49.6% were obese, and 32.1% had dyslipidaemia. These factors pose a high risk for cardiovascular disease, which can be controlled (modifiable) in the early years. From a public health perspective, one way to mitigate the risk of comorbidities is to address common modifiable behavioural risk factors [16].

Advancements in this century, such as artificial intelligence and machine learning, have attracted numerous researchers in the field of public health. Poudel et al. utilised machine learning methods to predict cognitive health in adults by considering various lifestyle and sociodemographic factors, along with the neighbourhood environment [17]. Their research proposed a machine learning-based regression approach, employing techniques such as gradient boosting machines, support vector machines, neural networks, and linear models.

Jang et al. [18] presented a deep learning-based decision support system for establishing and implementing public health and social measures during the COVID-19 pandemic. They proposed using Long Short-Term Memory (LSTM) neural networks for this purpose. Chadaga et al. [19] also utilised deep learning methods for diagnosing COVID-19 patients and proposed a decision support system incorporating explainable artificial intelligence techniques. Explainable artificial intelligence has also been employed to detect polycystic ovary syndrome (PCOS) [20]. Additionally, deep learning methods have been used for tuberculosis detection and classification from chest X-rays [21].

Haneef et al. [22] proposed a machine learning-based approach to estimate diabetes incidence in France using supervised machine learning techniques. They employed linear discriminant analysis (LDA), flexible discriminant analysis (FDA), and decision trees to estimate the occurrence of diabetes. In a nationwide study of the Danish population, diabetes prediction was performed using logistic ridge regression, classification was carried out using random forest, and a decision tree was validated using gradient boosting methods [8].

Lopes et al. [23] introduced an unsupervised machine learning approach for classifying the risk of preterm birth in Brazil, utilising municipality socio-economic data and the rate of preterm births. The data were clustered using three algorithms: K-means, principal component analysis, and density-based spatial clustering of applications with noise (DBSCAN).

As mentioned earlier, the studies highlight the application of advanced regression and machine learning models on extensive nationwide data. Logistic regression models utilise raw probabilities to predict outcomes, while neural network models employ numeric outputs with multiple layers and work with training, testing, and validation data. However, there is a gap in the literature regarding the prediction of multimorbidity using these models. Nevertheless, these models can assist in identifying and predicting risk factors for multimorbidity, a prevalent public health issue worldwide.

There is a lack of literature on predicting a risk assessment model for multimorbidities among the Saudi population. The Sharik Association for Health Research has developed the Sharik Health Indicators Surveillance System (SHISS), a nationwide, multi-wave, survey-based population health surveillance program initiated in early 2020 and conducted quarterly [24]. In this research, we utilised the dataset collected at different times in 2020 to estimate the health and risk factors in Saudi Arabia. Therefore, this study uses statistical and machine learning models to predict multimorbidity in the Saudi population.

## 2. Methodology

### 2.1. Research Design

The data are extracted from the “Sharik” Health Indicators Surveillance System (SHISS). This survey is a quarterly, cross-sectional, phone interview survey performed in all 13 administrative regions of KSA [24]. A trained data collector performed each interview, which lasted about 4 min. SHISS employs the ZdataCloud^®^. [25] research data collection system to manage the distribution of the sample and avoid human-related sampling bias. This system includes integrated eligibility and sampling modules. For the response to be successfully uploaded to the database, all questions are kept mandatory to be answered by the participants. The ZdataCloud^®^ database was used to code and store all the data.

### 2.2. SHISS Sampling and Sample Size

Saudi Arabia is divided into 13 administrative regions. The SHISS proportionate quota sampling method, stratifying by age, gender and region, was employed to obtain an equitable distribution of participants. Based on the Saudi Arabian median age of 36 years, SHISS divides its quota sample into two age categories (18–36 and 37+), resulting in 52 strata. The SHISS sample size was computed using a medium effect size of roughly 0.25, with 80 percent power of the test and 95 percent confidence level [11]. To produce a total of 6968 participants/wave, each quota required at least 134 individuals and a total sample of 536 per region.

### 2.3. SHISS Participants and Recruitment

Only residents of Saudi Arabia who spoke Arabic and were at least 18 years old were eligible to participate. To find possible volunteers, the Sharik Association for Research and Studies (Sharik) produced a random phone number list. Individuals that are interested in participating in future research projects make up the Sharik database. It has an increasing number of registered participants, with more than 180,000 spread out across Saudi Arabia’s 13 regions. On up to three occasions, participants were contacted via phone. In case of no response from the contacted ones, a new number was produced from the database with comparable demographics. This procedure was repeated until the quota was completed and closed automatically. Participants were assessed for eligibility by the interviewer based on the above-mentioned quota completion requirements after getting their consent to participate.

### 2.4. Study Variables

In this survey participants were asked about their demographic characteristics such as age, gender, per week vegetable and fruit consumption, weekly exercise (either moderate or intense), and the smoking status of the participants. They were further asked about presence of different morbidities such as diabetes, hypercholesteremia, hypertension, stroke, genetic disorder, cancer, chronic lung disease, asthma, and heart disease. Weight in kg and height in centimetres were recorded. A numeric variable labelled as body mass index (kg/m^2^) was generated. It is the ratio of weight in kg divided by height in m^2^. This measure is used to classify a person’s obesity level, which is a binary variable classifying each individuals into obese or non-obese using the cut-off value of BMI ≥ 30 kg/m^2^ as per the definition of Center of Disease Classification (CDC).

The presence of multimorbidity was defined as the person having any of the above mentioned two or more morbidities. The coding and ranges of the variables under study are given in Appendix A.

### 2.5. Data Analysis Tools

The analyses comprised three phases. First, univariate and bivariate analysis were performed to get an insight into the descriptive characteristics of the participants. In the second phase, a logistic regression model was employed, and in third stage, an artificial neural network (ANN) model was applied to the data. The accuracy of the models was checked at each stage. The description of the analysis in each phase was defined hereunder.

### 2.6. Descriptive Statistics

The qualitative variables such as gender, smoking status, and multimorbidity were presented with frequency and proportion. Quantitative variables such as age, per week vegetable and fruit consumption, and weekly intense and moderate physical activity were presented as the mean and standard deviation.

### 2.7. Risk Assessment Models

#### 2.7.1. Logistic Regression Model

As the survey data were based on the presence/absence of multimorbidity, the binary logistic regression model was used to assess the effect of independent variables on multimorbidity. Analyses were run in two stages: first, unadjusted models were built, i.e., studying the effect of individual independent variables on the presence of multimorbidity. Further, an adjusted model was built to determine the combined effect of all the independent variables on the presence of multimorbidity.

#### 2.7.2. Artificial Neural Network Model

Survey data on presence/absence of multimorbid conditions can be reduced to a two-dimensional matrix of n causes by n causes (‘n by n’ matrix) for each individual. When the multimorbidity profiles are grouped together, patterns emerge. This is a classification problem for which an artificial neural network (ANN) is suitable. ANN is a mathematical model that gets its inspiration from the human brain. The human brain consists of a network of interconnected biological neurons and similarly, ANN consists of interconnected processing units known as artificial neurons, and the pattern of connection of the neurons is called the architecture of the neural network. Over time, several neural network models have been developed, but the most widely used is the backpropagation neural network (BPNN) model. This is a layered model with an input layer (with *n* units), hidden layer (with *r* hidden units), and output layer (with *m* output units). In this model, an input pattern $\bar{x}=\left(x_{1},x_{2},\ldots x_{n}\right)$ is presented to the network. Each input $x_{i},\;\;where\;i=1,\;\ldots n$ is connected to every hidden unit $\;h_{j},\;\;where\;j=1,\;\ldots r$. Additionally, each hidden unit is connected to every one of the *m* output neurons [26].

Part of the procedure adopted in machine learning is to train the algorithm on a test set and use another set of data for the model validation and the last set of data for the application. The training algorithm comprises an iterative outer loop (called an epoch) and an iterative inner loop. In the backpropagation algorithm, the iterative inner loop (for each pattern) comprises four steps: (1) feed-forward computation (going from the input to the hidden layer to the output), (2) backpropagation to the output layer, (3) backpropagation to the hidden layer, and (4) updating of the weights [26,27].

The training algorithm of BPNN involves four basic steps: weight initialisation, calculation of the net output or activation of each neuron at each layer, and the backpropagation of errors. At the weight initialisation stage, weights are initialised with random values, and then these weights are used to produce the output by multiplying these weights with the inputs received at each layer level. The product sum is then calculated to produce the output of each neuron at each layer. The output of each neuron is feed-forwarded to the next layer until it finally reaches the final or the output layer. The output of final layer is then compared with the desired output to calculate the error. This error is then back-propagated to the previous layers and the weights are updated accordingly to adjust the error. Figure 1 shows the flowchart of the proposed scheme employing BPNN as a classifier.

## 3. Results

### 3.1. Descriptive Statistics

The data include the participation of 23,098 individuals. Table 1 compares the presence of multimorbidity across different independent variables. The results show that respondents with multimorbidity were older in age compared to respondents without any multimorbidity. About three quarters of males (76.2%) and a similar proportion among females (77.2%) were free of comorbidities. The difference between males and females was not significant. For all the health behavioural risk factors included, there was significant difference between the multimorbid and the non-multimorbid. This includes vegetable consumption per week, fruit consumption per week, intense physical activity per week, moderate physical activity per week, and smoker of any type.

### 3.2. Logistic Regression Model

Table 2 shows the odds ratio with the 95% confidence interval from the univariable (unadjusted) and multivariable (adjusted) logistic regression model for the predictors of multimorbidity. For the univariable model, the highest odds ratios were for smokers (1.29). After adjusting for the other variables in the model, smokers continued to have the highest odds ratios followed by the female sex. In the unadjusted model, the odds ratio for females was close to 1 (0.95), but this increased to 1.32 in the adjusted model.

The model accuracy was checked via the receiver operating characteristic (ROC) curve, as shown in Figure 2. The area under the curve was 0.77 (95% CI: 0.765–0.779). This indicates that the model is fitted relatively well to the data.

### 3.3. Backpropagation Neural Network

Using the same sets of variables as used in the logistic regression model, a three-layer BPNN was designed in Matlab to classify the collected data into two categories, i.e., comorbid and non-comorbid patients. For training, 70% of the total data were used, while 15% of the data were used for validation, and 15% were used for testing, respectively. Accuracy and other parameters such as sensitivity and specificity were calculated for training, validation, and testing by varying the number of neurons in the hidden layer. Table 3 shows the accuracy values for training, validation, and testing, respectively.

Table 4 further mentioned sensitivity, specificity, precision, the f-score, and AUC at different neurons.

The overall accuracy is the average of three accuracies (training, validation, and testing). However, to examine the performance of a trained neural network, testing accuracy is of prime importance in contrast to other accuracies. It can be seen from the table above that the maximum testing accuracy of 80.7% was obtained for 30 neurons in the hidden layer. However, in contrast, the best value of training accuracy was 79.5%, while the best value for validation accuracy was 79.6%. Cross-entropy was used as a loss function. Note that the lowest cross-entropy of 0.22398 was achieved with 30 hidden layer neurons in the BPNN, with the highest testing accuracy of 80.7 % at 60 epoch. This is shown in the Figure 3 below with the help of a dotted line.

Similarly, Figure 4 shows the training state curve of BPNN with 30 hidden neurons. The gradient of the training algorithm achieved its optimal (minimum) value of 0.0051097 at epoch 66, where further training of the network was stopped to freeze the weights of the neural network.

Figure 5 shows the Receiver Operating Characteristic (ROC) curves for training, validation, testing, and overall. The ROC curve is a characteristic plot between the true positive rate (TPR), also called sensitivity, to the false positive rate, also known as specificity. It is clear from the graphs for all the cases that the curves lie in the upper half of the plane. Therefore, the top left corner of the graph indicates a better classification capability of the classifier. The blue curve shows the ROC for multimorbidity, whereas the green one is for no- multimorbidity. It can be obviously seen in the testing ROC that the curves for the true positive rate and false positive rate lie far from the straight line toward the top left corner, therefore, showing better classification for the testing data.

Moreover, it is also clear from the confusion matrices, shown in Figure 6, that the sensitivity or TPR for testing data is 61.3%, for multimorbidity 83.3%, and for no-multimorbidity 80.7% for the testing data. The AUC values of training, validation, testing, and overall were 61.8%, 61.4%, 63.7%, and 62.1%, respectively.

Figure 7 shows a plot of testing accuracy versus the number of hidden layer neurons. It can be observed that increasing the neuron number in the hidden layer results in increasing the accuracy. However, once the number of neurons reached 30, it reached its maximum value. After that, it drops down and increasing the number of neurons does not increase the accuracy further. The graph shows a general decreasing trend in accuracy with an increase in the number of neurons.

Similarly, Figure 8 shows the number of epochs versus the testing accuracy. Maximum testing accuracy of the trained network is observed at 60 epochs.

## 4. Discussion

With the rapid increase in non-communicable diseases (NCDs) such as diabetes and hypertension, most Saudi health literature has focused on the comorbidities of these NCDs (as well as communicable diseases such as COVID-19) rather than multimorbidity in general [28,29,30]. One of the few exceptions is the study by Algabbanih et al. [31], which analysed multimorbidity in Saudi Arabia using Sharik data. Their study utilized pooled data from two national health surveys conducted in March 2018 and July 2018, based on the reported health status. Algabbanih et al. [31] included the following variables in assessing the risk factors for multimorbidity: age, gender, level of education, reported health status, smoking status, employment, marital status, and income. With the exception of education, at least one category was found to be significant for all these variables. The significant predictor categories for multimorbidity were: age over 45 years, being female, reported health status of ‘good’ or less, being retired or unable to work, being widowed, and a monthly income between SAR 10,000 to SAR 15,000. In our study, only three of these variables were included: age, gender, and smoking status. Additionally, we included ‘healthy behaviour variables’, namely, weekly vegetable consumption, weekly fruit consumption, intense physical activity per week, and moderate physical activity per week.

Akter et al. [32] conducted a meta-analysis of machine learning approaches to assess the increased risk of COVID-19. Their study also identified age and gender as the most significant predictors. Similar to Algabbanih et al. [31], female gender and smoking status were found to be significant predictors of multimorbidity. In our study, we did not categorize age as was performed by Algabbanih et al. [31]. The average age of individuals with multimorbidity in our study was 47.1 years. Since this age range does not fall under the category of ‘older adult’, the age variable was not a significant predictor of multimorbidity. It is worth noting the caution raised by the Australian Institute of Health and Welfare (AIHW) in their study of multimorbidity in Australia:

“It is important to note that while some characteristics, such as age, sex, or smoking status, may be more common in people with multimorbidity, it is not possible to say that multimorbidity is caused by these characteristics with the current data. Similarly, it is not possible to say that the characteristics examined are the result of having multimorbidity. For example, while being overweight or obese may be a risk factor for multimorbidity, it could also result from a person being limited in the physical activity they can participate in due to their multimorbidity.”[33]

Although there was a significant difference between the multimorbid and non-multimorbid groups in terms of ‘healthy behaviour variables,’ they were not significant predictors of multimorbidity in the adjusted model, as all their odds ratios were close to 1.

The unadjusted logistic regression model helped us determine the effect of individual variables on multimorbidity. The unadjusted estimate can be implemented in further research if the researcher has different sets of variables. For instance, only three covariates were shared between our study and the study by Algabbanih et al. [31]. Hence, the results are comparable based on unadjusted estimates. However, the adjusted analysis helped us determine the best and most effective predictors for the outcome variable, i.e., multimorbidity. As discussed earlier, smoking was the most influential predictor for multimorbidity.

The overall accuracy of the logistic regression model was 77%. The machine learning model used in the paper slightly improved this accuracy to 80.7%. Further research can explore other machine learning methods for predicting multimorbidity.

In this paper, BPNN was used as a tool to classify patients as morbid or non-morbid. The results indicate that machine learning approaches can be used to classify patients into different groups associated with various data attributes. Furthermore, while the case study presented in this paper focused on a binary classification problem, machine learning approaches can also be applied to solve multiclass problems. Various algorithms, including supervised and unsupervised learning methods, can be used in future research. The potential use of machine learning in the field of public health can lead to the development of decision support systems that assist health experts in making decisions related to different public health issues and policies. In the case of high-dimensional data, dimensionality reduction techniques such as PCA or ranking methods based on statistical approaches like ANOVA, combined with other machine learning algorithms, can be utilised for such decision support systems.

This study is the first to predict factors associated with multimorbidity using nationwide data and applying machine learning methods. One limitation of this study is the limited number of variables included, which makes it difficult to compare the results with the study by Algabbanih et al. [31]. It is hoped that this study will stimulate further Saudi research on the vital topic of multimorbidity.

## 5. Conclusions

The study predicted factors affecting multimorbidity among the Saudi population using nationwide data. Both statistical and machine learning models produced closed outcomes with relatively high accuracy. Females and smokers were found to be more prone to multimorbidity. Increasing age and fruit consumption also played a significant role in multimorbidity. Multimorbidity is a major concern among the Saudi population, and policymakers strive to highlight factors to the Saudi population to reduce this rampant issue. The significant variables identified in our study can serve as a spotlight for policymakers in Saudi Arabia to reduce the risk of multimorbidity among the population.

The proposed approach has applications to many real-world problems. For example, in actuarial practices, the proposed model can be used to reduce the complexity of developing an actuarial model that predicts the comorbidity of patients, which is a labour- and skill-intensive task. The proposed approach can also be helpful for medical practitioners in classifying patient data for decision-making and support.

However, it should be noted that the variables used in the models were limited. Applying machine learning models to these nationwide Saudi data allows future researchers to predict multimorbidity. Further evidence-based research can strengthen our study findings.

## Figures and Tables

**Figure 1 healthcare-11-02176-f001:**
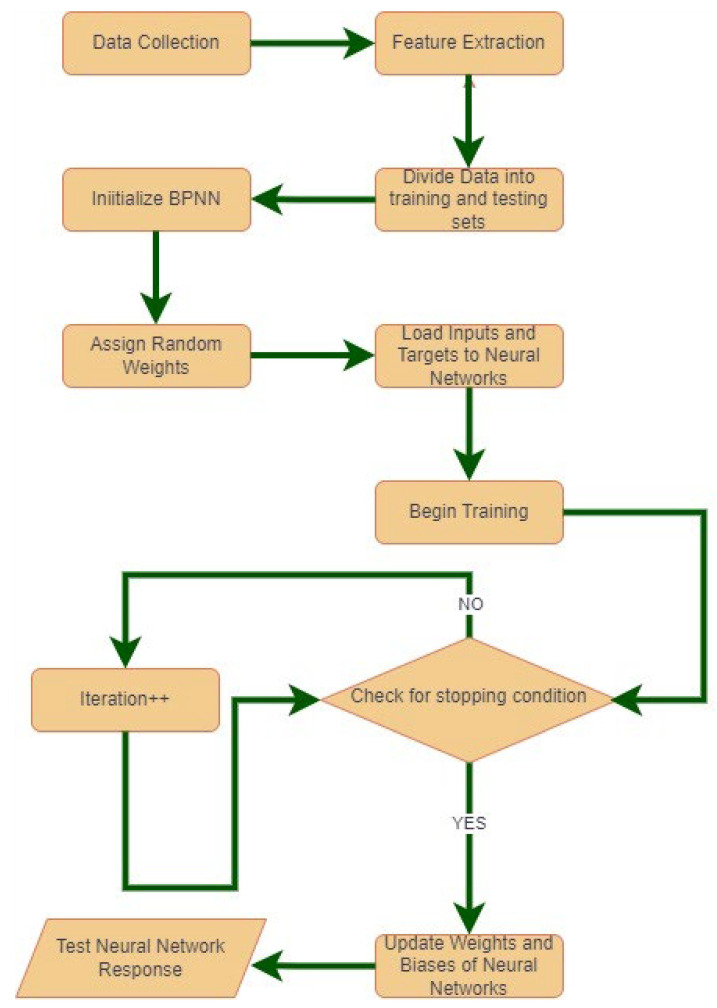
Flowchart of the proposed scheme.

**Figure 2 healthcare-11-02176-f002:**
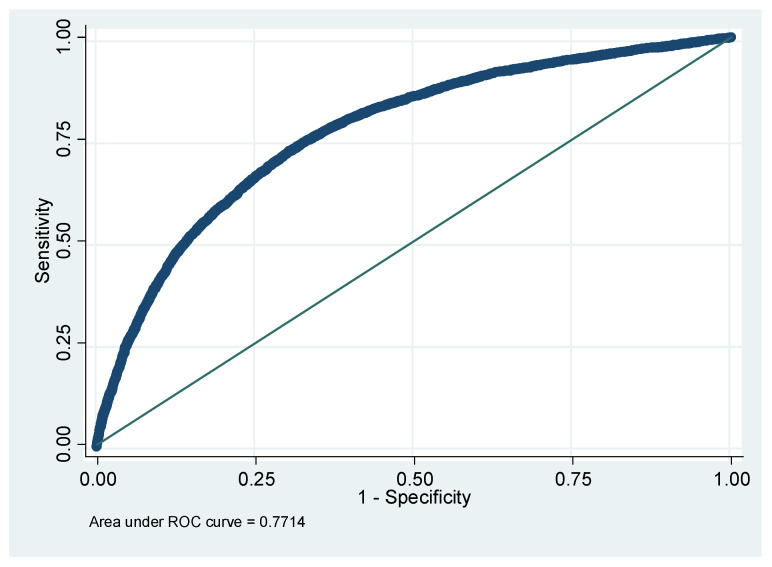
Receiver operating curve (ROC) for the multivariable logistic regression model.

**Figure 3 healthcare-11-02176-f003:**
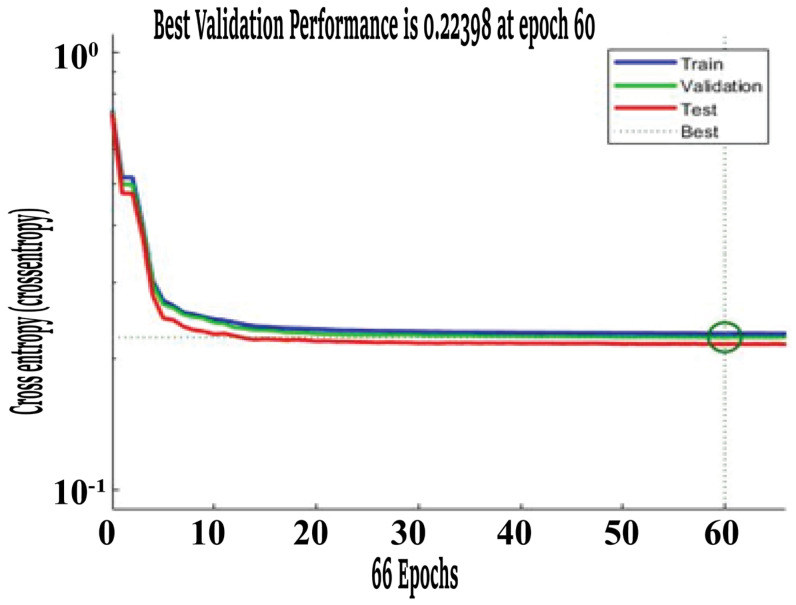
BPNN performance curve with 30 hidden layer neurons.

**Figure 4 healthcare-11-02176-f004:**
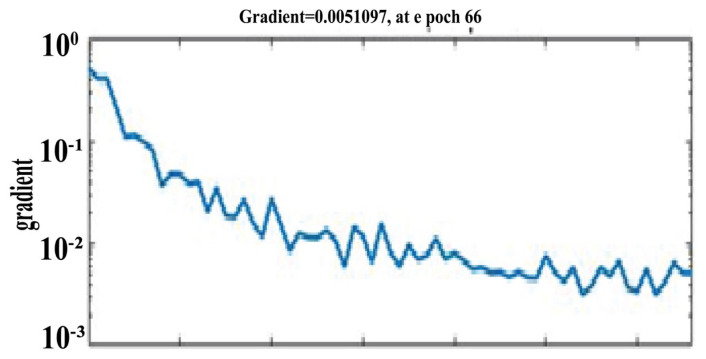
Training state curve of BPNN with 30 neurons.

**Figure 5 healthcare-11-02176-f005:**
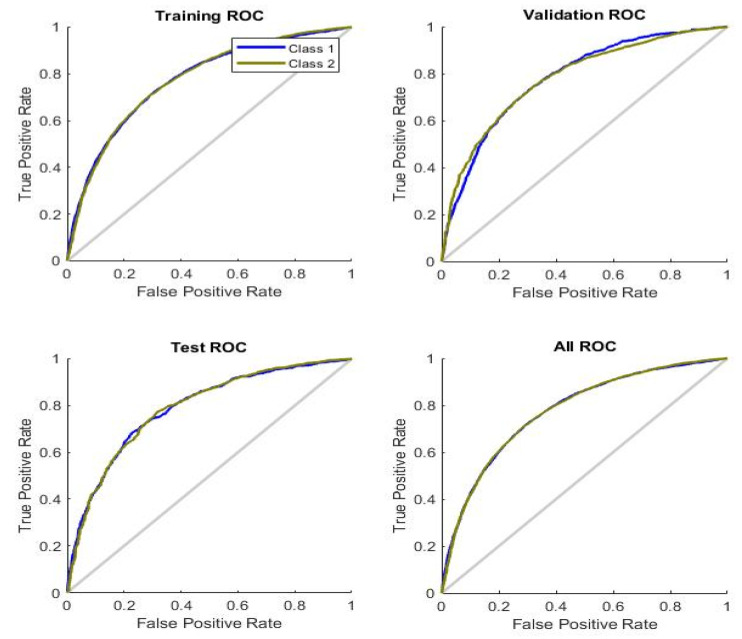
ROC curves of the BPNN with 30 neurons.

**Figure 6 healthcare-11-02176-f006:**
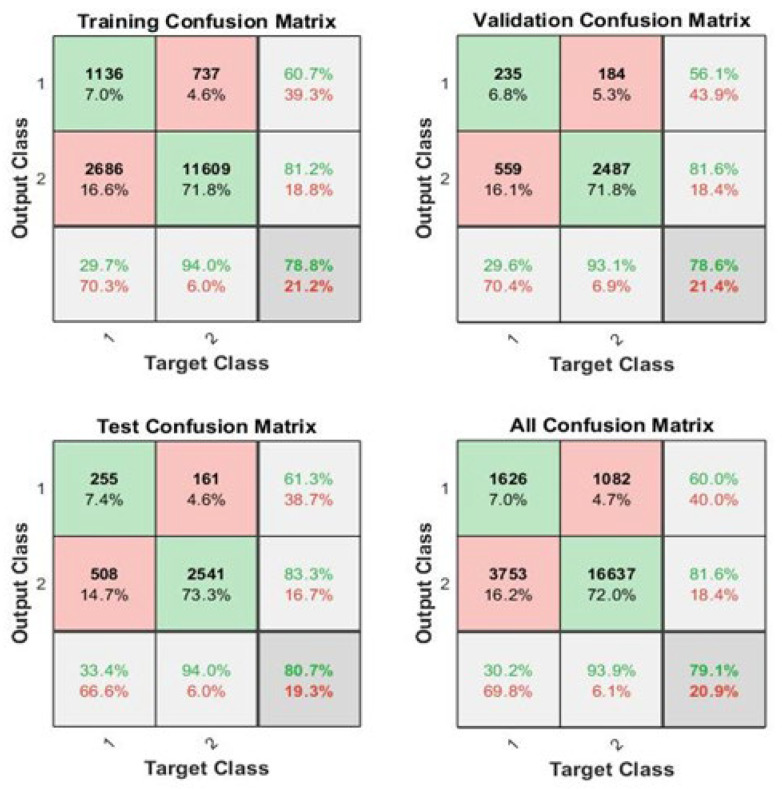
Confusion matrices of the BPNN with 30 neurons. The diagonals green cells highlight true positive or negative values, the off-diagonals pink cells highlight false positive or negative values. The silver cells highlight marginal proportions. The grey cells highlight grand proportion.

**Figure 7 healthcare-11-02176-f007:**
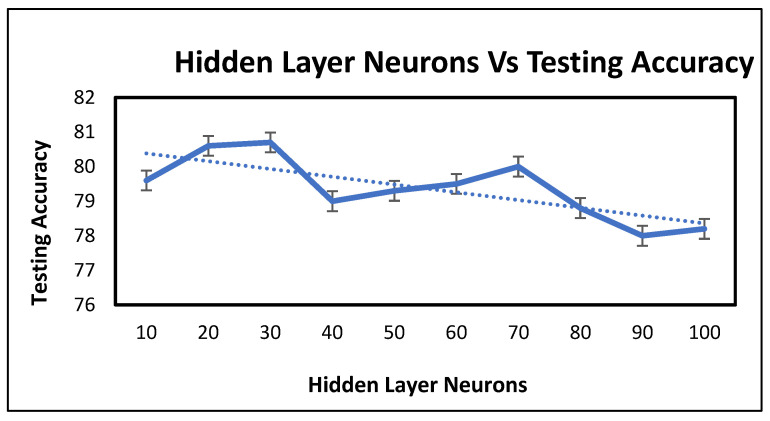
Hidden layer neurons of the BPNN vs. testing accuracy.

**Figure 8 healthcare-11-02176-f008:**
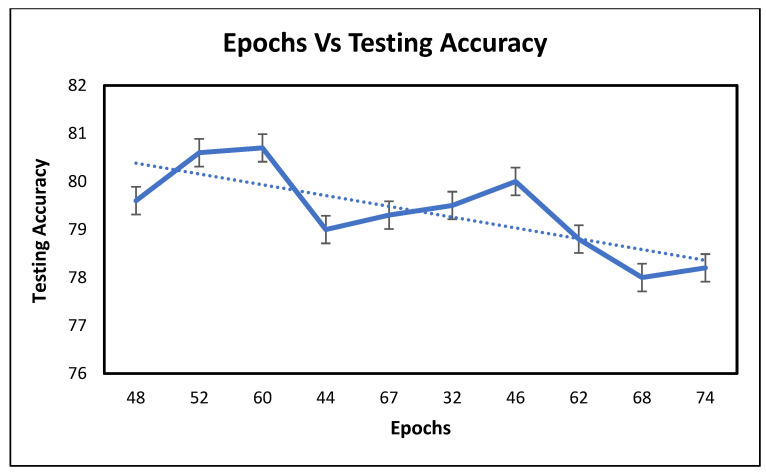
No. of epochs vs. testing accuracy.

**Table 1 healthcare-11-02176-t001:** Demographic characteristics and presence of multimorbidity.

		Presence of Multimorbidity		*p*-Value
		No	Yes	
Age (years)		33.8 (12.2)	47.1 (14.2)	**<0.0001**
Gender	Male	8717 (76.2%)	2716 (23.8%)	0.096
	Female	9002 (77.2%)	2663 (22.8%)	
Vegetable consumption per week (#)		3.4 (2.1)	3.6 (2.1)	**<0.0001**
Fruit consumption per week (#)		2.7 (2.1)	3.1 (2.2)	**<0.0001**
Intense physical activity per week (#days)		1.8 (2.2)	1.4 (2.0)	**<0.0001**
Moderate physical activity per week (# days)		2.3 (2.4)	2.1 (2.3)	**<0.0001**
Smoker of any type	No	12,966 (78.0%)	3653 (22.0%)	**<0.001**
	Yes	4753 (73.4%)	1726 (26.6%)	

Significant results are marked in bold.

**Table 2 healthcare-11-02176-t002:** Univariable and multivariable logistic model for the predictors of multimorbidity.

	UOR (95% CI)	AOR (95% CI)
Age (years)	1.07 (1.07–1.08)	1.08 (1.07–1.08)
Females	0.95 (0.89–1.01)	1.32 (1.23–1.43)
Vegetable consumption per week (#)	1.06 (1.05–1.08)	0.99 (0.97–1.01)
Fruit consumption per week (#)	1.09 (1.08–1.11)	1.05 (1.03–1.06)
Intense physical activity per week (#days)	0.92 (0.91–0.94)	0.97 (0.96–0.99)
Moderate physical activity per week (#days)	0.95 (0.94–0.97)	0.98 (0.96–0.99)
Smoker	1.29 (1.21–1.38)	1.74 (1.61–1.89)

UOR: univariable odds ratio; AOR: adjusted odds ratio.

**Table 3 healthcare-11-02176-t003:** Accuracy values for training, validation, and testing.

Input Neurons	Hidden Layer Neurons	Output Neurons	Training Accuracy (%)	Cross Entropy	Epoch	Validation Accuracy (%)	Testing Accuracy (%)	Overall Accuracy (%)
8	10	2	79.1	0.23198	48	78.4	79.6	79.1
8	20	2	79.1	0.23591	52	77.7	80.6	79.1
8	30	2	78.8	0.22398	60	78.6	80.7	79.1
8	40	2	79	0.22725	44	79.6	79	79.1
8	50	2	79.2	0.22652	67	78.6	79.3	79.1
8	60	2	78.7	0.22704	32	79.5	79.5	78.9
8	70	2	79.3	0.23009	46	78.6	80	79.3
8	80	2	79.3	0.23128	62	79.2	78.8	79.2
8	90	2	79.5	0.22509	68	79.3	78	79.2
8	100	2	79.5	0.23008	74	79	78.2	79.2

**Table 4 healthcare-11-02176-t004:** Accuracy measures for training, validation, testing, and overall.

No.	No. of Neurons	Sensitivity (Recall)				Specificity				Precision				F-Score				AUC			
		Training	Validation	Testing	Overall	Training	Validation	Testing	Overall	Training	Validation	Testing	Overall	Training	Validation	Testing	Overall	Training	Validation	Testing	Overall
1	10	94	93.9	93.5	94	30.3	26	31.9	29.9	81.5	81	82.3	81.5	87.3	87.0	87.5	87.3	62.2	60.0	62.7	62.0
2	20	94.1	94	94.7	94.2	30.1	26.7	29.8	29.5	81.5	80	83	81.5	87.3	86.4	88.5	87.4	62.1	60.4	62.3	61.9
3	30	94	93.1	94	93.9	29.7	29.6	33.4	30.2	81.2	81.6	83.3	81.6	87.1	87.0	88.3	87.3	61.9	61.4	63.7	62.1
4	40	94.9	95	94.9	94.9	26.4	29.3	27.3	27	81	81.4	81	81.1	87.4	87.7	87.4	87.5	60.7	62.2	61.1	61.0
5	50	94.9	94.6	94.5	94.8	27.7	26.3	26.6	27.3	81.1	80.8	81.7	81.1	87.5	87.2	87.6	87.4	61.3	60.5	60.6	61.1
6	60	94.6	95.6	95.3	94.9	26.4	27.9	24.5	26.4	80.8	80.9	81.4	80.9	87.2	87.6	87.8	87.3	60.5	61.8	59.9	60.7
7	70	94.9	94.7	95.1	94.9	28.4	24.1	29.5	27.9	81.2	80.8	81.8	81.3	87.5	87.2	88.0	87.6	61.7	59.4	62.3	61.4
8	80	94.1	94.4	93.8	94.1	31	28.2	29.4	30.3	81.7	81.5	81.5	81.6	87.5	87.5	87.2	87.4	62.6	61.3	61.6	62.2
9	90	94.6	94.2	94.1	94.5	29.3	30.5	29.5	28.9	81.6	81.6	80.4	81.4	87.6	87.4	86.7	87.5	62.0	62.4	61.8	61.7
10	100	95	94.6	94.8	94.9	27.8	27.3	27.3	27.7	81.5	81.1	80	81.2	87.7	87.3	86.8	87.5	61.4	61.0	61.1	61.3

## Data Availability

Data are available upon request to the contact author.

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
