# Peer review of "Predicting Multimorbidity Using Saudi Health Indicators (Sharik) Nationwide Data: Statistical and Machine Learning Approach"

_healthcare, 2023, doi:10.3390/healthcare11152176_

Round 1
Reviewer 1 Report
The authors presented a study to predict the factors for multimorbidity in the Saudi population using statistical and machine learning approaches. The study made use of an available nationwide database. While it is a crucial study primarily for the Saudi population, the article has some issues as listed below.
- Redundant language is used throughout the paper, which impacts its readability. For compactness and clarity of content, it is highly recommended to edit the paper.
- Sharik Health Indicators Surveillance System (SHISS) is first mentioned at the end of the introduction section. Is there any other study in the literature that makes use of this database? This should be clearly discussed at this point itself. This point is also necessary to highlight the presented study's novelty, which is currently not obvious.
- In section 2.7, the adopted models are not clearly described. For instance, for logistic regression, how do adjusted and unadjusted models differ from an implementation point of view? Please elaborate.
- For the artificial neural network in section 2.7.2, it would be better to provide an image to clearly describe the n causes by n causes matrix derived from the survey data.
- Please mention the units of various parameters in Table 1.
- Figure 5: Area under the ROC curves (AUROC) should also be mentioned and compared.
- Discuss more the observation stated in lines 292-293. Why increasing the number of neurons does not increase the accuracy further?
- Lines 320-323: The average age of 47.1 years for those with multimorbidity is similar to the age predicted in Algabbanih et al. (2019). Aren't these sentences contradictory then? Please explain.
Language needs to be improved at multiple places in the paper. The authors are strongly advised to go through the paper again and edit the language wherever required. For instance, there is a lot of redundancy in
- the abstract - It is not clear. Particularly, lines 28-29 are already mentioned earlier.
- the introduction. It can be compacted. For instance, 'comorbidity and multimorbidity' are repeated multiple times. It can be replaced with 'in this field' in line 42.
- Line 47: doesn't it include all the countries? if yes, this sentence can be compacted.
- Line 56: again redundant.
- Line 160: correct the typo. ' ... each individuals into obese or non-obese'
- Line 203: correct the spelling error. ' each pattern'
Reviewer 2 Report
I appreciate the opportunity to review this paper. In this paper the authors present a Machine learning application to predict multi-morbidity risk. They compare the performance of a neural network model to the traditional logistic regression model.
The paper is written in an easy to understand language, it provides a good amount of background in the introduction section and presents the methods used in an ordered and clear way. My greatest critique is that it is difficult to link the findings here to the healthcare aspect. The logistic regression model shows effects that are easy to be interpreted but that same interpretation is not offered for the neural network model. There is a lot of effort made to showcase the neural network’s performance but all that is useless from the healthcare point of view. Considering that the neural network model had a 80.7% fit vs a 77% for the logistic regression, the improvement may not be relevant to warrant the additional effort of using the neural network model. What did the neural network approach add?
I have also some additional concerns that may be important to address.
· The SHISS only allows citizens of Saudi Arabia to participate, why is that? KSA has a large population of non-citizens living in the country, why are they excluded? In the supplementary the nationality is coded, was it or was not considered?
· Why do we need to predict multi-morbidities if we can just ask the people interviewed what are their conditions?
· What is the total N?
· The table 1 needs description of the units and adding an additional column showing the values for the entire dataset would be useful.
· In the conclusion, an argument is made about the predictors “these variables can be a spotlight for the policy makers in Saudi Arabic to reduce the risk of multi-morbidity among Saudi populations” How?
The paper has some spelling and grammatical errors that need to be corrected.
Reviewer 3 Report
Dear authors,
Predicting multimorbidities is an an interesting idea. Please make the changes and resubmit.
Abstract: Please mention the meaning of multimorbidity in the abstract. Otherwise abstract is very good.
Introduction: Mention the research gaps and contributions of yours must be mentioned in points. This makes it easier for readers to understand novelty.
Literature review is very weak.. you need to add a minumum of 7 to 10 years.. ML is used as a decision support system for any diseases other than diabetes. I hope the following articles can be of help to you.
1.Chadaga K, Prabhu S, Bhat V, Sampathila N, Umakanth S, Chadaga R. A Decision Support System for Diagnosis of COVID-19 from Non-COVID-19 Influenza-like Illness Using Explainable Artificial Intelligence. Bioengineering. 2023 Mar 31;10(4):439.
2. Khanna VV, Chadaga K, Sampathila N, Prabhu S, Bhandage V, Hegde GK. A distinctive explainable machine learning framework for detection of polycystic ovary syndrome. Applied System Innovation. 2023 Feb 23;6(2):32.
3. Acharya V, Dhiman G, Prakasha K, Bahadur P, Choraria A, Prabhu S, Chadaga K, Viriyasitavat W, Kautish S. AI-assisted tuberculosis detection and classification from chest X-rays using a deep learning normalization-free network model. Computational Intelligence and Neuroscience. 2022 Oct 3;2022.
2. Methodology
Fig 1 is boring. Could you use an application called draw.io? You can make it colorful. It will also help you in the future.
Table 3: What is the optimizer used? loss function? activation function?
Is it possible to add a table and calculate precision, recall and f1-score and accuracy. All of these metrics can be easily calculated using the confusion matrix.
Please mention why deep learning performed better than logistic regression
Limitations and future work needs to be a special section
Add a threat to validation section
Conclusion is too small. Add atleast two paragraphs..
Entire paper must be proof read to check for minor issues and typos
Overall verdict: The paper is very novel, the data is novel and the results are good enough. I have no hesitation in accepting this wonderful paper after the authors address the minor issues.
Best of luck!
Good.
Round 2
Reviewer 2 Report
The authors made a great effort in improving the paper. The quality of the most current draft is noticeable.
I believe the authors resolved well my concerns although I have a single comment that I believe it should be addressed.
I believe there is some bias in the way SHISS samples the population. The authors report 95.2% of nationals responding the survey however, Saudi Arabia has a 20-30% non-nationals, it appears the sampling excludes a large chunk of the population. I suggest a disclaimer is added to the paper, either as a limitation or as some rationale for not adding these people in the methods.
I endorse this paper for publication provided that this concern is addressed.